# Design of a Novel Compact Bandpass Filter Based on Low-Cost Through-Silicon-Via Technology

**DOI:** 10.3390/mi14061251

**Published:** 2023-06-14

**Authors:** Hai Dong, Yingtao Ding, Han Wang, Xingling Pan, Mingrui Zhou, Ziyue Zhang

**Affiliations:** 1School of Integrated Circuits and Electronics, Beijing Institute of Technology, Beijing 100081, China; 3420225025@bit.edu.cn (H.D.); ytd@bit.edu.cn (Y.D.); 3220215100@bit.edu.cn (H.W.); panxingling0705@163.com (X.P.); linghuaibu@163.com (M.Z.); 2BIT Chongqing Institute of Microelectronics and Microsystems, Chongqing 400030, China

**Keywords:** compact bandpass filter, integrated passive device (IPD), miniaturization, three-dimensional (3D) integration, through-silicon-via (TSV)

## Abstract

Three-dimensional (3D) integration based on through-silicon-via (TSV) technology provides a solution to the miniaturization of electronic systems. In this paper, novel integrated passive devices (IPDs) including capacitor, inductor, and bandpass filter are designed by employing TSV structures. For lower manufacturing costs, polyimide (PI) liners are used in the TSVs. The influences of structural parameters of TSVs on the electrical performance of the TSV-based capacitor and inductor are individually evaluated. Moreover, with the topologies of capacitor and inductor elements, a compact third-order Butterworth bandpass filter with a central frequency of 2.4 GHz is developed, and the footprint is only 0.814 mm × 0.444 mm. The simulated 3-dB bandwidth of the filter is 410 MHz, and the fraction bandwidth (FBW) is 17%. Besides, the in-band insertion loss is less than 2.63 dB, and the return loss in the passband is better than 11.4 dB, showing good RF performance. Furthermore, as the filter is fully formed by identical TSVs, it not only features a simple architecture and low cost, but also provides a promising idea for facilitating the system integration and layout camouflaging of radio frequency (RF) devices.

## 1. Introduction

As one of the key components in radio frequency (RF) front ends and microwave systems, bandpass filters (BPFs) are extensively investigated and have spurred great advances in terms of structures, processes, and performance over the past decades. Several technologies such as low temperature co-fired ceramic (LTCC) [1,2], microstrip [3,4,5,6], substrate integrated waveguide (SIW) [7,8,9], and micro-electromechanical systems (MEMS) [10,11], are utilized to develop BPFs with specific characteristics such as broad bandwidth, high selectivity, low insertion loss, and compact size. Recently, with the rapid development of wireless communication, Internet of Things (IoT), and Artificial Intelligence (AI), the demands for miniaturization and integration of RF components are especially urgent, therefore, it is necessary to design filters that are more compact and easier to be integrated [12,13,14,15].

To achieve a smaller footprint, multi-layer structure and discriminating coupling are involved in the design of compact LTCC BPFs [16,17,18]. For SIW filters, some modified topologies such as the half-mode SIW (HMSIW) [19], and the ridge half-mode SIW (RHMSIW) [20,21], are proposed to optimize the device size. Besides, based on the spoof surface plasmon polaritons (SSPPs) in the microwave frequencies, several novel compact wideband SSPP filters [22,23] and hybrid SIW-SSPP filters [24,25] are developed for plasmonic systems, microwave circuits, and wearable devices. MEMS technology offers another alternative to the design of compact integrated passive devices (IPDs) [26,27], and the RF MEMS devices are superior in aspects of tuning, switching, and integration with other chips [28,29]. However, it is difficult for most of the above technologies to further shrink the footprints of filters to satisfy the increasing demands for the miniaturization of future electronic systems.

Three-dimensional (3D) integration and interposer technologies have emerged as effective solutions to the heterogeneous integration of multiple devices and chips [30,31,32]. As the core of 3D integration and interposer technologies, through-silicon-vias (TSVs) can vertically transmit electrical signals among different device layers, enabling high-performance systems within more compact areas [33,34,35]. Moreover, TSVs can also constitute several functional devices, such as capacitors [36] and inductors [37,38]. For RF applications, glass substrate is also used due to its high electrical resistivity and low electrical loss, where through-glass-vias (TGVs) are developed similar to TSVs [39,40]. Some of the reported literature has used TSV/TGV based capacitors together with inductors fabricated by traditional integrated circuit (IC) processes (or vice versa) to form BPF structures [41,42,43,44], showing great potential to improve the compactness of filters. To further simplify the device structure and reduce the process complexity, a compact, fully TGV-based BPF is proposed in [45]. Nevertheless, the manufacture of TGVs is not as mature as that of TSVs, which is more flexible and reliable in fabrication processes such as via etching and metallization. On the other hand, TSV technology has been widely studied and has achieved various progresses, for example, some low dielectric constant polymers are developed as substitutes for conventional SiO_2_ liners, which feature low parasitic capacitance, good insulation quality, and low fabrication cost and complexity [46,47,48]. In this paper, novel and compact IPD structures including capacitor, inductor, and BPF are designed based on TSV technology. High resistivity silicon (HRS) with a resistivity of 1 × 10^4^ Ω·cm is used as the substrate has higher RF performance, as well as a better capability to be integrated with other Si based devices. Besides, polyimide (PI) is applied as the TSV liners, which can offer good electrical performance and be easily fabricated by a low-cost and simple technique named vacuum-assisted spin coating [49,50,51]. The electrical characteristics of the TSV based coaxial-like capacitors and solenoid inductors are discussed in detail. Notably, the utilized TSVs are identical and thus can be fabricated in a single batch. Employing the TSV based capacitor and inductor elements, a compact BPF structure is designed and evaluated, which possesses several attractive advantages, particularly with respect to a small footprint, simple structure, and low cost.

This paper is organized as follows: Section 2 and Section 3 discuss the influences of design parameters on the performance of the TSV based coaxial-like capacitors and the TSV based solenoid inductors, respectively. Section 4 describes the modeling and characterization of the proposed BPF. Finally, the conclusions are summarized in Section 5.

## 2. TSV Based Coaxial-like Capacitor

Traditional on-chip capacitors are usually planar metal-insulator-metal (MIM) structures occupying large device areas. By sequentially depositing multiple layers via holes, 3D MIM capacitors named through-silicon-capacitors (TSCs) are fabricated [52,53,54]. Even though the TSCs have a smaller footprint than traditional planar capacitors, their fabrication processes are relatively complex. Actually, the TSV structure is a natural metal-insulator-semiconductor (MIS) capacitor, although the capacitance is always regarded as an unwanted parasitic parameter. Besides, if one TSV is placed at the center of several annularly arranged TSVs that are connected together, a coaxial-like capacitor structure can be formed, where the central conductor of the inner TSV and the central conductors of the surrounding TSVs act as the two terminals of the capacitor [55], respectively.

Based on the low-cost PI-TSV technology, a topological coaxial-like capacitor structure is firstly proposed, and the basic capacitor element comprised of five identical TSVs is shown in Figure 1. The HRS substrate is hidden for better observation. The TSVs have a diameter of 30 μm and a height of 300 μm, the thickness of PI liner is 0.5 μm, and the pitch between inner and outer TSVs is $25\sqrt{2}$ μm as the pitch between adjacent outer TSVs is set to be 50 μm. It can be seen that four surrounding TSVs are connected to a port by a backside Cu redistribution layer (RDL), and the central TSV is linked to the other port by a frontside RDL. Actually, more surrounding TSVs contributes to increasing the capacitance of the coaxial-like capacitor [55], however, to build capacitors that have specific capacitances, the number of surrounding TSVs is set to be four, forming a square shape that can be easily duplicated.

Considering the sandwiched substrate between the inner and surrounding TSVs, the capacitance of such a coaxial-like capacitor is formed by the series connection of the parasitic capacitances of the inner TSV, the capacitance of the HRS substrate, and the parasitic capacitances of the surrounding TSVs. Therefore, the total capacitance of the capacitor element is influenced by the structural parameters of the TSVs and the pitch between central and outer TSVs.

Figure 2 plots the simulated capacitance–frequency characteristics of the capacitor element in Figure 1 under various design parameters, including TSV height, TSV diameter, thickness of PI liner, and pitch between inner and outer TSVs. As shown in Figure 2a,b, the increases in TSV height and diameter enlarge the capacitances over the simulated frequencies, due to the increased equivalent terminal areas in the capacitor. In addition, with the decreases in PI thickness and pitch between inner and outer TSVs, the parasitic capacitances of the PI-TSVs and the capacitance of the substrate are increased, respectively, leading to the increase in the total capacitance.

Moreover, arrays of coaxial-like capacitors with larger capacitances can be easily formed by the duplication and parallel connection of the basic square capacitor elements in both X- and Y-directions, which can be described as the matrix topology of coaxial-like capacitors.

## 3. TSV Based Solenoid Inductor

Three-dimensional (3D) solenoid inductors comprised of TSVs/TGVs and RDLs have been proposed for a smaller footprint and higher inductance density than traditional planar spiral structures [39,40]. To further reduce the manufacturing cost and complexity, a new 3D solenoid inductor structure based on low-cost PI-TSVs and corresponding RDLs is designed, and the basic inductor element with two turns is shown in Figure 3. The physical parameters of the PI-TSVs are the same as those of the coaxial-like capacitor in Section 2. It is shown that two X-directional adjacent TSVs and the RDLs form an equivalent coil, and the equivalent cross-sectional area of the inductor is decided by the pitch between the two TSVs and the TSV height. Actually, 3D solenoid inductors can store magnetic field energy more efficiently than planar structures, thus achieving larger quality factors within smaller footprints [39]. On the other hand, the pitch between two adjacent Y-directional TSVs acts as the loop pitch of the inductor, thus influencing the performance of the inductor together with the cross-sectional area and the number of turns. Therefore, the impact of various structural parameters including TSV height, TSV diameter, thickness of PI liner, and pitches between two adjacent TSVs in X- and Y-directions on the inductance and quality factor of the inductor element is analyzed, and the simulated results are shown in Figure 4 and Figure 5.

Figure 4a shows that the inductance increases with TSV height as the cross-sectional area of the inductor increases. Besides, as the frequency-dependent alternating current (AC) resistance also increases with TSV height, the variation trend of the quality factor is determined by the weights of the increasing ratios of the reactance and the resistance at various frequencies. Therefore, the quality factor increases with TSV height at lower frequencies but decreases at higher frequencies, as shown in Figure 5a. As the increase in TSV diameter decreases the effective cross-sectional area, the inductance is declined accordingly, as shown in Figure 4b. However, as the resistance is decreased more significantly, the quality factor is increased instead, as shown in Figure 5b. Figure 4c and Figure 5c illustrate that the thickness of PI liner has little impact on the inductance and quality factor of the inductor element, which is because both the effective cross-sectional area and the resistance are almost unchanged, and the reduction in energy loss with thicker liner is not significant. As shown in Figure 4d, the influence of the pitch between two X-directional TSVs on the inductance is similar to that of the TSV height as both of them directly decide the effective cross-sectional area of the inductor. Notably, as the pitch between the two TSVs has less impact on the resistance of the coil compared to the TSV height, the quality factor is increased with the inductance, as shown in Figure 5d. Finally, the pitch between two Y-directional adjacent TSVs is actually influencing the equivalent length of the inductor, therefore, the inductance is slightly decreased with larger pitch, as shown in Figure 4e. On the other hand, as the inductive coupling is also decreased due to proximity effect, the quality factor is nearly the same, as shown in Figure 5e.

Similar to the coaxial-like capacitor in Section 2, the basic solenoid inductor element can also constitute inductors with larger inductances in a simple topological method. As the proposed inductor element is composed of repeatable loops of identical TSVs and RDLs, an array of 2 × N TSVs can easily be formed to multiply the inductance, where N refers to the number of turns. Besides, the pitches between two adjacent TSVs in X- and Y-directions can also be adjusted to tune the inductance of the basic inductor element.

## 4. TSV Based Bandpass Filter

Based on the proposed coaxial-like capacitor and solenoid inductor structures, a novel BPF structure is designed, which is fully comprised of identical low-cost PI-TSVs. A typical prototype of third-order Butterworth BPF is adopted and the simplified circuit model of the designed filter with a central frequency (f_0_) of 2.4 GHz is shown in Figure 6. Three capacitors and three inductors are used in the filter, and the parameter values of the components are derived and validated by the Keysight ADS circuit simulator. To obtain PI-TSV based capacitors and inductors of specific values, the coaxial-like capacitor element and the solenoid inductor element are topologically evolved.

Figure 7 shows the designed PI-TSV based BPF structure that is assembled by three arrays of coaxial-like capacitors and three 3D solenoid inductors with multi-turns. Notably, all TSVs in the filter are identical thus can be fabricated simultaneously, which greatly reduces the manufacturing cost and complexity. Moreover, as shown in Figure 7b, the footprint of the compact BPF is only 0.814 mm × 0.444 mm excluding the surrounding test framework, corresponding to an ultra-small electrical dimension of about 0.0065 λ_0_ × 0.0035 λ_0_. 

The transmission characteristics of the proposed BPF are analyzed using full-wave simulations in Ansys HFSS, and the return loss (S_11_) and insertion loss (S_21_) curves are plotted in Figure 8. Besides, the S_11_ and S_21_ curves obtained from the theoretical circuit model are also shown in Figure 8 for comparison. According to the simulated results, the f_0_ of the BPF is 2.4 GHz, the 3-dB bandwidth is 410 MHz, and the fraction bandwidth (FBW) is 17%. In addition, the in-band insertion loss is less than 2.63 dB, and the return loss in the passband is better than 11.4 dB, showing good RF performance. Note that there are some deviations between the S-parameter curves of the circuit model and the physical model of the proposed PI-TSV based filter, which can be attributed to the parasitic parameters introduced by the TSV and RDL structures.

Table 1 presents the comparisons of the designed PI-TSV based BPF with other published works in respect of RF characteristics, device sizes, technologies, and manufacturing costs and complexities. It is shown that the proposed compact PI-TSV based BPF has comparable RF performance with the state-of-the-art designs, while it occupies a smaller footprint than most of the reported ones. Even though that the filter in [44] has an even smaller size, its fabrication flow is relatively complex as it stacks several parallel plate capacitors on the TGV based inductors. Moreover, as the proposed filter in this work is developed based on a single Si interposer with a thickness of 300 μm and is fully formed by low-cost PI-TSV structures and corresponding RDLs, it is able to be fabricated by the relatively mature TSV technology without the need to introduce extra bonding steps or IC processes. Meanwhile, all TSVs can be synchronously fabricated within one process round, which further reduces the manufacture cost and complexity. Besides, the PI-TSV based BPF is also attractive in the modern RF/electronic micro-systems as it is compatible with 3D integration technology and can be easily integrated with other chips and modules.

## 5. Conclusions

In this work, fully PI-TSV based IPDs including capacitor, inductor, and BPF are designed and evaluated. The impacts of various design parameters on the performance of the TSV based coaxial-like capacitor and the TSV based solenoid inductor are analyzed, providing a guidance to the development of such components. Capacitors and inductors with specific values of capacitances and inductances can be easily achieved by the topologies of the basic coaxial-like capacitor element and solenoid inductor element. Based on this topological method, a novel compact BPF structure is proposed, which is composed of several arrays of coaxial-like capacitors and multi-turns solenoid inductors. The 3-dB bandwidth and the fraction bandwidth (FBW) of the filter is 410 MHz and 17%, respectively. Besides, the in-band insertion loss is less than 2.63 dB, and the return loss in the passband is better than 11.4 dB, showing good RF performance. Notably, the device size of the BPF is only 0.814 mm × 0.444 mm (0.0065 λ_0_ × 0.0035 λ_0_). Moreover, as the proposed BPF is fully formed by low-cost PI-TSVs with the same structural parameters and corresponding RDLs, it features lower manufacturing cost and complexity compared to other reported designs. Furthermore, the BPF is convenient to be integrated with other devices as it is built in a natural Si interposer structure. As a conclusion, the proposed design of compact BPF is promising for the miniaturization and integration of modern RF applications.

## Figures and Tables

**Figure 1 micromachines-14-01251-f001:**
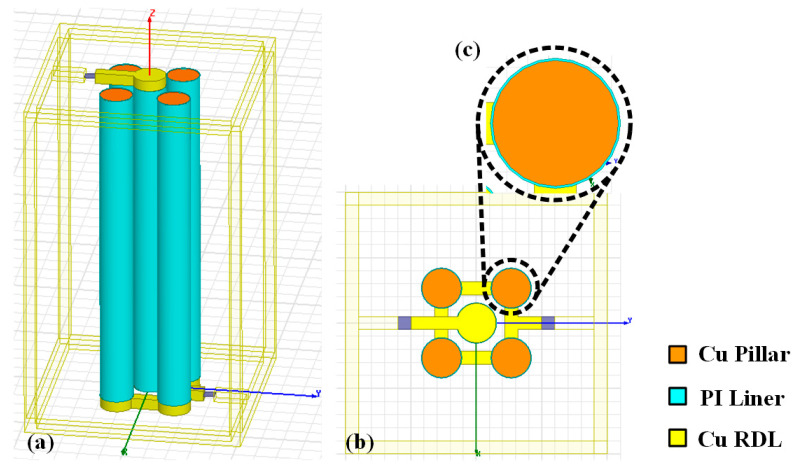
Basic capacitor element for the PI-TSV based coaxial-like capacitor structure: (**a**) side view, (**b**) top view, (**c**) is the enlarged image of a single TSV in (**b**).

**Figure 2 micromachines-14-01251-f002:**
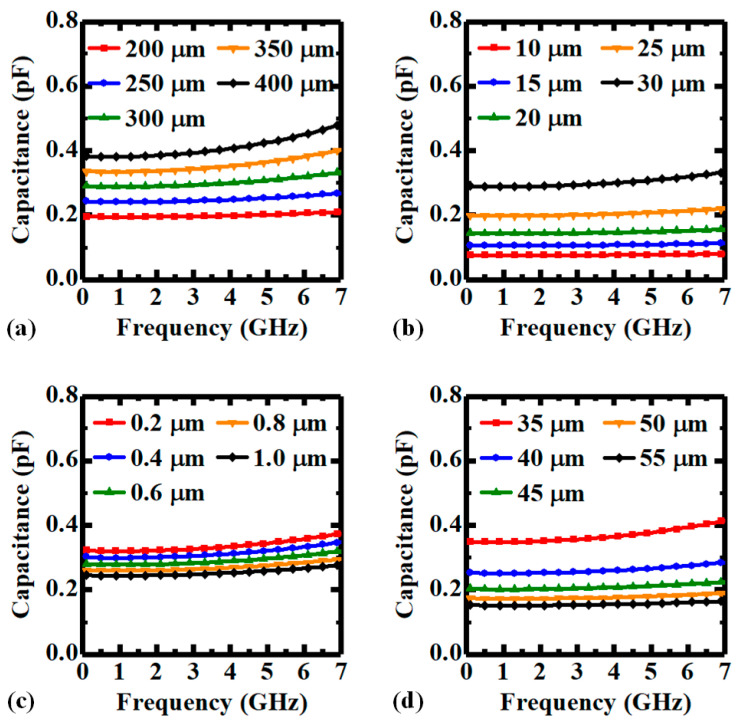
Capacitance–frequency characteristics of the PI-TSV based capacitor element with various (**a**) TSV heights, (**b**) TSV diameters, (**c**) thicknesses of PI liner, and (**d**) pitches between inner and outer TSVs.

**Figure 3 micromachines-14-01251-f003:**
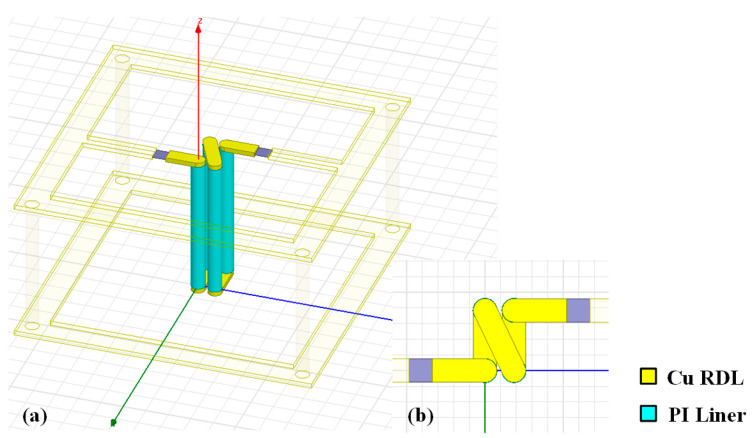
Basic inductor element for the PI-TSV based solenoid inductor structure: (**a**) side view, (**b**) top view.

**Figure 4 micromachines-14-01251-f004:**
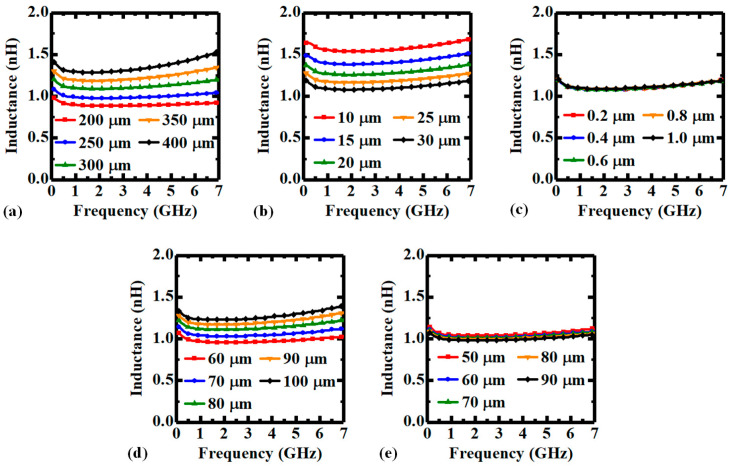
Inductance–frequency characteristics of the PI-TSV based inductor element with various (**a**) TSV heights, (**b**) TSV diameters, (**c**) thicknesses of PI liner, (**d**) pitch between two X-directional adjacent TSVs, and (**e**) pitch between two Y-directional adjacent TSVs.

**Figure 5 micromachines-14-01251-f005:**
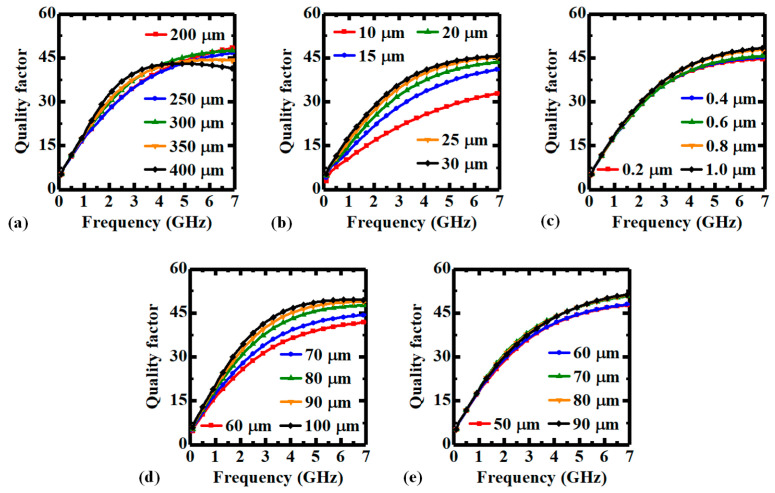
Quality factor-frequency characteristics of the PI-TSV based inductor element with various (**a**) TSV heights, (**b**) TSV diameters, (**c**) thicknesses of PI liner, (**d**) pitch between two X-directional adjacent TSVs, and (**e**) pitch between two Y-directional adjacent TSVs.

**Figure 6 micromachines-14-01251-f006:**
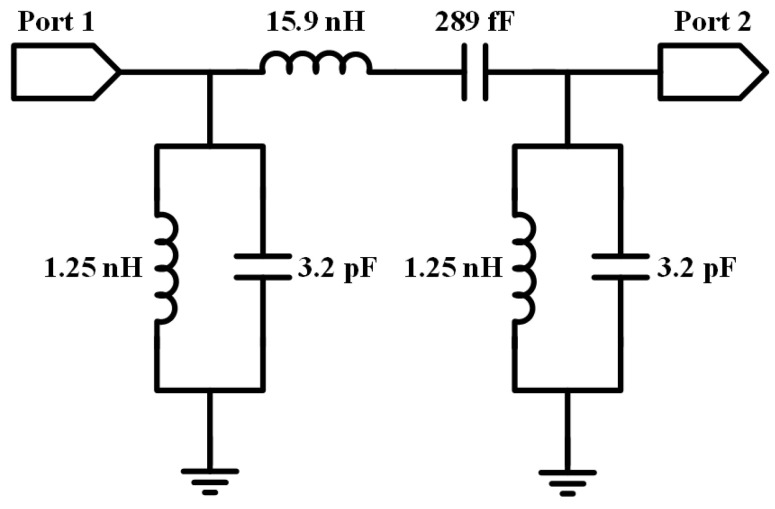
Circuit model of the proposed third-order Butterworth BPF.

**Figure 7 micromachines-14-01251-f007:**
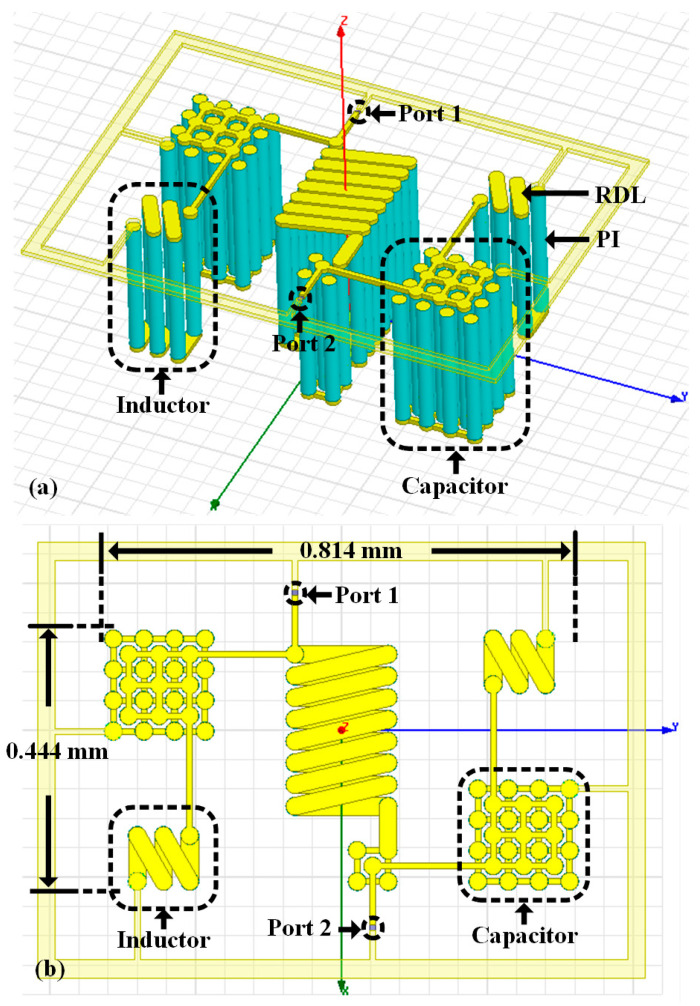
Structure of the proposed PI-TSV based BPF: (**a**) side view, (**b**) top view. A coaxial-like capacitor component and a solenoid inductor component are indicated. The device footprint is 0.814 mm × 0.444 mm.

**Figure 8 micromachines-14-01251-f008:**
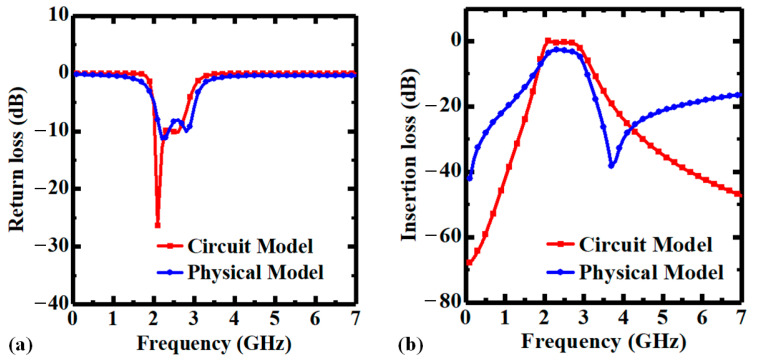
Transmission characteristics of the proposed PI-TSV based BPF: (**a**) return loss (S_11_), (**b**) insertion loss (S_21_). The red lines are obtained from the ideal circuit model, and the blue lines are simulated results of the physical model in HFSS.

**Table 1 micromachines-14-01251-t001:** Comparisons of the proposed BPF with other published designs.

References	f_0_ (GHz)	FBW (%)	Insertion Loss (dB)	Return Loss (dB)	Size (mm × mm)	Technology
[17]	2.4	12.5	2.4	15	2.63 × 2.61	LTCC
[18]	2.6	10.2	2.47	20	2 × 1.7	LTCC
[23]	2.4	174	1.1	11	87.5 × 12.8	SSPP
[43]	2.65	62.9	0.6	30	1 × 0.5	TGV and IC
[44]	2.45	31.5	2.4	15	0.44 × 0.33	TGV and IC
[56]	2.7	2	3.8	14	44.8 × 16	DPAM *
[57]	2.33	39.8	0.65	15	18.5 × 18.5	Micro-strip
[58]	2.4	10.4	0.87	15	9.4 × 23.1	Micro-strip
[59]	2.4	12.1	1.2	15	19.1 × 11.2	Micro-strip
This work	2.4	17	2.63	11.4	0.81 × 0.44	PI-TSV

* DPAM: Direct print additive manufacturing.

## Data Availability

Not applicable.

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
