# Peer review of "Design of a Novel Compact Bandpass Filter Based on Low-Cost Through-Silicon-Via Technology"

_micromachines, 2023, doi:10.3390/mi14061251_

Round 1

Reviewer 1 Report

Design of a Novel Compact Bandpass Filter Based on Low-Cost Through-Silicon-Via Technology is proposed. The following are observations/suggestions/modifications:

1.       The abstract should have all the filter statistics, please ensure to include as many as possible.

2.       The introduction is adequately drafted. Please ensure to have the presence of all the latest articles.

3.       Page 2 Line 82 & 83: Can you please rewrite the lines for better understanding?

4.       Is there any reasons why the dimension of TSV is specified as 25 root 2, if yes then mention it. Else final dimensions may be included in the manuscript.

5.       Figure 1: The text may be kept away from the image for better visibility.

6.       Page 4 Line 126: The quotes may not be required for matrix topology.  Figure 5 is adequately drawn.

7.       Page 6 Line 158: Besides, the frequency-dependent… Please re-write for better understanding. Figure 7 illustration is very nice. Figure 8 image size may be increased for better readability.

8.       Page 8 Line 201: Can you possibly include the electrical dimensions in addition to the mechanical dimensions?

9.       Page 8 Line 211: Please check subscript for resonance frequency

10.   The comparison is adequately carried out.

11.   Conclusion may also have electrical dimensions. References are adequate.

Minor improvements are required in typo/grammar.

Reviewer 2 Report

The authors designed novel integrated passive devices (IPDs) including capacitor, inductor, and bandpass filter by employing TSV structures.  The filter is fully formed by identical TSVs, it not only features a simple architecture and low cost, but also provides a promising idea for facilitating the system integration and layout camouflaging of radio frequency (RF) devices.

The work is interesting but needs minor revision. Some of the points are as follows.

1 Figure 2. Capacitance-frequency characteristics of the PI-TSV based capacitor element with various (a) TSV heights, (b) TSV diameters, (c) thicknesses of PI liner, and (d) pitches between inner and outer TSVs.     There are TSV height, TSV diameter, thickness of PI liner, and pitch between inner and outer TSVs. However, step sizes are ignored.

2 Table 1 presents the comparisons of the designed PI-TSV based BPF with other published works in respect of RF characteristics, device sizes, technologies, and manufacturing costs and complexities. The authors claimed It is shown that the proposed compact PI-TSV based BPF has comparable RF performance with the state-of-the-art designs, while it occupies a smaller footprint than most of the reported ones.   How about Reference  [44] ?  Add more explanation.

.  

3Figure 8. Transmission characteristics of the proposed PI-TSV based BPF: (a) return loss (S11), (b) insertion loss (S21). The red lines are obtained from the ideal circuit model, and the blue lines are simulated by HFSS.  There are red lines and blue lines.  Blue lines are called This work. We think it is not appropriate as to Ideal   for comparison.  Simulation is recommended

Minor editing of English language required

Round 2

Reviewer 1 Report

The authors have incorporated the suggestions.

Grammar to be rechecked.